# Early Stage In Vitro Bioprofiling of Potential Low-Molecular-Weight Organoboron Compounds for Boron Neutron Capture Therapy (BNCT)—Proposal for a Guide

**DOI:** 10.3390/cells13100798

**Published:** 2024-05-08

**Authors:** Zbigniew J. Leśnikowski, Filip Ekholm, Narayan S. Hosmane, Martin Kellert, Eiji Matsuura, Hiroyuki Nakamura, Agnieszka B. Olejniczak, Luigi Panza, Louis M. Rendina, Wolfgang A. G. Sauerwein

**Affiliations:** 1Laboratory of Medicinal Chemistry, Institute of Medical Biology PAS, Lodowa 106, 93-232 Lodz, Poland; 2Department of Chemistry, University of Helsinki, P.O. Box 55, FI-00014 Helsinki, Finland; filip.ekholm@helsinki.fi; 3Department of Chemistry and Biochemistry, Northern Illinois University, DeKalb, IL 60115, USA; hosmane@niu.edu; 4Deutsche Gesellschaft für Bor-Neutroneneinfangtherapie DGBNCT e.V., University Hospital Essen, 45122 Essen, Germany; martin.kellert@outlook.de (M.K.); luigi.panza@uniupo.it (L.P.); 5Graduate School of Interdisciplinary Science and Engineering in Health Systems, Okayama University, Okayama 700-0005, Japan; eijimatu.01@gmail.com; 6Laboratory for Chemistry and Life Science, Institute of Innovative Research, Tokyo Institute of Technology, Yokohama 226-8501, Japan; hiro@res.titech.ac.jp; 7Screening Laboratory, Institute of Medical Biology PAS, 106 Lodowa, 93-232 Lodz, Poland; aolejniczak@cbm.pan.pl; 8Dipartimento di Scienze del Farmaco, Università degli Studi del Piemonte Orientale “A. Avogadro”, L.go Donegani, 2/3-28100 Novara, Italy; 9School of Chemistry, The University of Sydney, Sydney, NSW 2006, Australia; louis.rendina@sydney.edu.au; 10Department of Radiation Oncology, University Hospital Essen, University Duisburg-Essen, 45122 Essen, Germany

**Keywords:** BNCT, boron carriers, bioprofiling, in vitro methods, pre-clinical testing, drug development

## Abstract

Given the renewed interest in boron neutron capture therapy (BNCT) and the intensified search for improved boron carriers, as well as the difficulties of coherently comparing the carriers described so far, it seems necessary to define a basic set of assays and standardized methods to be used in the early stages of boron carrier development in vitro. The selection of assays and corresponding methods is based on the practical experience of the authors and is certainly not exhaustive, but open to discussion. The proposed tests/characteristics: Solubility, lipophilicity, stability, cytotoxicity, and cellular uptake apply to both low molecular weight (up to 500 Da) and high molecular weight (5000 Da and more) boron carriers. However, the specific methods have been selected primarily for low molecular weight boron carriers; in the case of high molecular weight compounds, some of the methods may need to be adapted.

## 1. Introduction

Boron Neutron Capture Therapy (BNCT) is a promising, binary modality for treating certain types of cancer based on its extraordinary ability to act selectively only on cells that have accumulated sufficient amounts of boron-10 (^10^B) atoms. BNCT is based on the high cross-section of the non-radioactive isotope ^10^B to capture thermal neutrons. The immediately occurring nuclear reaction ^10^B(n,α)^7^Li produces two charged particles, an alpha particle and a lithium nucleus. The path lengths of these particles are up to 10 µm resulting in an energy deposition limited to the diameter of a single cell [1]. This enables the selective irradiation of tumor cells with a high amount of ^10^B, while normal cells in the immediate vicinity, possessing a lower boron content, are spared. In the past, nuclear research reactors were essential for generating low-energy neutron fields with the intensity required for this type of therapy. This hindered the widespread use of this technology for decades. Recently, mainly due to the development of neutron sources based on accelerators that can be installed in hospitals, interest in this technology has sharply risen again worldwide [2,3]. Successful BNCT requires two basic components: a neutron source with suitable properties and a drug that is enriched with the non-radioactive isotope ^10^B, that can preferentially accumulate in tumor tissue. In this context, it is also worth paying attention to a new, emerging technology, Neutron Capture Enhanced Particle Therapy (NCEPT). NCEPT offers BNCT without an external neutron source and is based on ^10^B capturing thermal neutrons produced inside the treatment volume during irradiation with proton and carbon ions [4,5].

The growing interest in BNCT is leading to the emergence of BNCT centers around the world [6], creating a rising demand for more sophisticated boron carriers. In the past, various approaches and a range of molecules have been investigated to introduce boron into tumor cells. They include low molecular-weight-carriers such as amino acids, nucleic acid precursors, DNA binding molecules, and porphyrin derivatives, but also high-molecular-weight-carriers such as monoclonal antibodies or monoclonal antibody fragments, high boron- loaded DNA-oligomers, dendrimers, dextrans, polylysine, etc.

The substances used in BNCT boron administration must meet several essential criteria: high ^10^B enrichments (99.6% is currently achieved, the natural isotope composition is around 20%); no systemic toxicity; selective capture at the level of the tumor; tumor/normal tissue ratio higher than 3-4:1; a tumor/blood concentrations ratio of at least 3:1; efficient boron carrier molecule transporters; absolute boron content in tumor at least ~20 µg ^10^B/g per tumor (or ~10^9^ atoms/cell); rapid blood and tissue clearance that should nonetheless persist in the tumor cell throughout the irradiation [7]. Concerning the first criterion, it is worth noting that although ^10^B enrichment being as high as possible is desirable in principle, a balance should be maintained between the costs and benefits achieved. An increase in benefits from 99% to 99.6% enrichment does not justify the cost for the higher ^10^B contents. Furthermore, more selective boron carriers may be able to ensure that enough ^10^B is accumulated in tumor tissues even at lower levels of ^10^B enrichment. However, some recent developments in the ^10^B market have raised concerns that a sustainable supply of highly enriched ^10^B is not always guaranteed. Since the ^10^B concentration in the compound to be used is an essential part of its effect, the use of a secured ^10^B concentration is more important than the optimal enrichment of 99.6%. Regarding the requirement to accumulate ∼10^9 10^B atoms per cancer cell, it is worth recalling that this value has been used for many years assuming that the boron compound is evenly distributed throughout the cytoplasm. If boron agents were developed to target subcellular structures such as DNA, the mitochondrial membrane, etc., they would probably require lower concentrations than these.

Boron carriers for BNCT have been developed since the 1950s, and during this time, thousands of different “potential boron carriers” were synthesized and described, but only a few of them have been deeply studied for applications in BNCT. More concerningly, these compounds were tested with a different set of assays and methods, and even if the methods were similar, the research protocols were often different. These differences and the lack of standardization make it fundamentally difficult to compare the properties of published boron carriers and make judicious decisions about the selection of a molecule for further in-depth research.

In search of better boron carriers than the BPA and BSH used to date [8], new preparations are constantly being proposed, such as the recently described boronotyrosine [9] or pteroyl-*closo*-dodecaborate (PBC-IP) conjugated with a 4-(p-iodophenyl)butyric acid moiety [10], as well as the meta isomer of L-BPA, (L)-3-dihydroxy-borylphenylalanine or 3-BPA [11] that has a better solubility as compared to L-BPA and may result interesting innovative formulations. Lists of other candidates for boron compounds for BNCT can be found in a recently published article by Oloo et al. [12] and at the website of the German BNCT Society DGBNCT [13]. Special attention merits the need to develop a theranostics approach for BNCT [14]. One can be sure that the search for new boron carriers will continue and accelerate in both industrial and academic laboratories.

It now seems to be the right time to adopt a guideline for a set of uniform, basic, pre-clinical experimental tests that should be performed on compounds published as “potential boron compounds” for BNCT. A good example of an attempt to standardize approaches to new drug research for BNCT is the recently published reference to protocols for in vitro and in vivo evaluation methods of BNCT boron drugs [15]. In their recommendations, the authors briefly discuss methods for evaluating boron carriers both in vitro and in vivo. The methods for assessing the amount of boron both in cell cultures and in blood and tissues were discussed in detail. In the proposed in vitro set of tests, the authors considered the determination of cytotoxicity, cellular uptake, uptake mechanism, and subcellular distribution. The proposed basic in vivo tests include acute toxicity, boron uptake in tumors and various organs, and distribution of boron compounds in tissues. BNCT in animal models and the evaluation of anti-tumor effects by neutron irradiation was also included in the recommended test battery [15].

We assume that the structural characterization of new delivery agents, in terms of validating their integrity and purity, is the responsibility of the researchers who are going to evaluate it further. In the guidelines proposed here, we limit the proposal to the first, basic stage of evaluation of new boron compounds, by restricting the proposal to the in vitro phase. At the same time, we propose specific methods and protocols for these initial tests to facilitate standardization. Such an early-stage in vitro profiling program should be robust, stripped down, and focused on identifying a promising hit compound. We also believe that at this stage of screening studies of potential boron carriers, it is not necessary to investigate the mechanism of cellular uptake and subcellular distribution. A detailed understanding of the mechanism of action is not worth the money and time at this initial stage of boron carriers development.

In the early phase of drug discovery, when many compounds are being developed and optimized, the basic parameters listed below are often detected using high-throughput methods. However, in the case of molecules containing boron, libraries suitable for high-throughput screening (HTS) are not yet available. This concerns molecules with a single boron atom as well as derivatives with boron clusters. Therefore, the methods we propose are suitable for low/medium-throughput assays for small collections of several or tens of compounds.

We propose the following tests (see Figure 1):solubility measurement in H_2_O and estimation of solubility in DMSO (soluble or insoluble).pK_a_, and log P/D determination.stability of the compounds at pH 1, 7.4, 9, and human plasma.cytotoxicity in vitro, in human glioblastoma multiforme cells U87MG, and/or squamous cell carcinoma, SAS, related to head and neck cancer or A375 for melanoma, as an example of cancer cells, and in HEK293 as an example of “normal” tissue cells. Of course, there is no obstacle to determining cytotoxicity in a larger number of cell lines, but these four should always be taken into account. As a criterion of cytotoxicity, we propose a concentration that reduces the number of viable cells by 50% (CC_50_) compared to untreated cells, lower than 100 mM.cellular uptake measured by ICP-AES in U87MG and HEK293c ells used in cytotoxicity studies.

## 2. Protocols for Testing BNCT Compounds

### 2.1. Principles

#### 2.1.1. Solubility in H_2_O

Solubility in water, the main component of biological fluids, is a key physicochemical property for the success of any drug candidate. It determines the compound absorption, bioavailability, and safety. The assessment of solubility in dimethyl sulfoxide (DMSO) is, in turn, needed because it is a standard solvent used to prepare stock solutions that are then used in most physicochemical and biological in vitro assays, even if it is not inert to cells. It is therefore recommended that the final DMSO concentration in biological tests should not exceed 0.5% and preferably 0.1%. Although many calculation methods have been developed for predicting water solubility from the structure of a compound, generally showing reasonable accuracy, they are not suitable for use in the case of boron-containing compounds, especially boron clusters. We suggest no less than 1 mM as the required minimum solubility of the compound in water (molar solubility), and a minimum of 100 mM in DMSO. The 1 mM value for water solubility is within the range defined as “slightly soluble”. This solubility, although low, allows for the testing of boron compounds and does not result in the rejection of too many derivatives at the beginning of the boron compound evaluation. For comparison, the solubility of BPA which is considered highly insoluble in water, is 2.8–3.3 mM (0.6–0.7 mg/mL [9]).

Among various experimental methods for determining the water solubility of drugs [16,17,18], we propose the nephelometric method and measurement in 96- or 364-well plates due to its versatility and applicability to measurements in the microscale. An example procedure is shown below in the experimental section.

#### 2.1.2. pK_a_ Determination

The ability to rapidly measure absorption properties (solubility, pK_a_, log P) concurrent to activity and transport provides a data-based molecular property assessment allowing promising compounds to quickly pass into exploratory development and, conversely, undesirable compounds will quickly fail.

As the majority of drugs are weak acids and/or bases, knowledge of the dissociation constant in each case helps in understanding the ionic form a molecule will take across a range of pH values. This is particularly important in physiological systems where the ionization state will affect the rate at which the compound can diffuse across membranes and obstacles such as the blood-brain barrier (BBB). The pK_a_ of a drug influences lipophilicity, solubility, protein binding, and permeability, which in turn directly affects pharmacokinetic characteristics such as absorption, distribution, metabolism, and excretion (ADME) [19].

There are currently several known methods of determining a pK_a_ value, namely: potentiometric titration, spectrometry, fluorometry, NMR, HPLC, conductometry, electrophoresis, voltammetry, solubility, partition coefficient, calorimetry, computational, and surface tension. Some of these techniques are more widely utilized and well-established compared to others [20]. The two main methods for determining the pK_a_ of a compound are potentiometric titration and spectrophotometric titration. The main advantage of the second method is the possibility to obtain a titration curve that allows an estimate to be made at any point without the need for an experiment. Spectrometric titration can be performed using a standard UV-Vis spectrophotometer (Thermo Fisher Scientific, Waltham, MA, USA) or one of the available automated systems. An example of the procedure can be found below in the experimental section.

#### 2.1.3. Log P/D Determination

Knowing the lipophilicity of a compound, which is usually measured by the log P/D value, helps to identify compounds that are more likely to be well absorbed and distributed in the human body. It should be noted that there are a considerable number of routes for the absorption of drugs across membranes, with transport by passive diffusion being the most common. To be absorbed by this route, drugs must be lipophilic enough to penetrate lipid membranes, but not so lipophilic that they become stuck there. Lipophilicity, the measure of a drug’s affinity for a lipid environment, has become a very important parameter, as it indicates the relationship between drugs and their biological, pharmacokinetic, and metabolic properties. Log D is a log of partition of all forms of the compound between the lipid and aqueous phases at specific pH. Log P is equivalent to log D for non-ionizable compounds and it represents the partition of the neutral form for ionizable compounds. The most popular mimic of the lipid phase is octanol

Lipophilicity can be measured by determining the distribution of a drug between an organic solvent, generally, *n*-octanol saturated with water (or buffer), and an aqueous (buffer) phase. The partition coefficient (P) refers to the ratio of compound concentrations in each phase and can be determined experimentally by a variety of methods including the well-known shake-flask method, potentiometric methods, chromatographic methods, and others [21]. We have found that the potentiometric method and spectrometric technique [22,23,24] performed with the Pion SiriusT3 system are particularly useful for the log D determination of ionizable compounds [25]. However, due to its popularity, general applicability, and gold standard status, we include an example of a shake flask procedure in the experimental section below.

#### 2.1.4. Evaluation of a Compound’s Stability at Various pH Levels and in Human Plasma

Evaluating the stability of drug substances and products is of great importance for the determination of drug quality. Both physical and chemical degradation reactions influence the stability of medicinal products. Chemical degradation processes include hydrolysis, oxidation, decarboxylation, elimination, isomerization, dimerization, epimerization, photodegradation, and dehydration. In the initial phase of the development of new biologically active molecules, most tests are carried out in vitro, in aqueous media. It is therefore particularly important to determine the stability of compounds in an aquatic environment. The resulting degradation can confuse results of the structure-activity relationship (SAR) and lead to false conclusions.

Hydrolysis tests can be carried out in buffers of different pH levels, and buffers with pH levels 1.0, 7.4, and 9.0 corresponding to the pH of the stomach, blood, and colon, respectively are often used. Testing the resistance of compounds to hydrolysis in buffers with different pH levels is necessary, but it is worth this supplementing with stability tests of the compounds in biological fluids, which are required at further stages of evaluation. As the minimum required stability of the compound at pH 7.4, we recommend not less than 90% after 24 h, and in blood plasma no less than 50% after 24 h. An example procedure for testing the stability of compounds in human plasma is shown in the experimental section below [26,27]. A simple procedure for assessing the stability of compounds in buffers of different pH levels, using LC-MS, can be found, e.g., in the Waters Co. application note [28].

#### 2.1.5. Cellular Toxicity

Each BNCT treatment requires the administration of a large amount of a ^10^B-containing compound to the patient undergoing therapy. During a typical BNCT session, the patient receives an intravenous infusion of 250–500 mg/kg body weight of the drug such as ^10^[B]BPA-fructose complex (BPA-F) over 2–3 h.

This requires that the boron carrier is non-toxic or has only minimal systemic toxicity. Short- and long-term toxicity studies of high doses of the drug in vivo can usually only be carried out in the later phases of pre-clinical and then clinical trials of the most promising drug candidates, once a lot of time and money has already been invested. It is therefore particularly important to assess the toxicity of the tested derivatives at an early stage. This is usually performed through in vitro tests of cellular toxicity in cell cultures. Such studies give no guarantee that a compound with low toxicity in vitro is also sufficiently non-toxic in vivo, but they make possible to eliminate compounds that are definitely toxic at an early stage.

There is considerable confusion in the literature regarding cytotoxicity studies with potential boron carriers, both in terms of the cell lines used, the concentrations of the compounds tested, and the duration of cell treatments with the drug.

We propose that each potential boron carrier be tested in two stages. In the first stage, toxicity would only be tested at a specific concentration of 0.1 mM. Compounds would be considered toxic if the percent viability value is 50% or less, and would be considered non-toxic if the percent viability value is >50% compared to the control (cells without treatment). Compounds that are non-toxic according to this criterion are subjected to further tests in the concentration range of 0.01–10 mM to determine the CC_50_ value, i.e., the concentration at which the viability of the cells decreases by 50%. As a criterion for cytotoxicity, we suggest assuming a CC_50_ value of at least 1 mM for a compound classified as non-toxic. For comparison, the cytotoxicity of BSH and BPA was assessed as CC_50_ = 3.79 mM in V79 cells [29] and CC_50_ > 2 mM in B16 cell line [30], respectively.

As basic lines for preliminary cellular cytotoxicity tests, it is advisable to choose 2 or 3 tumor cell lines, which show good colony formation potentials, good tumorigenic potential in mouse models, without showing elevated multidrug resistance (MDR) pumps expression. For BNCT experiments, SAS, U87MG, and A375 can be recommended for oral cancer, high grade glioma, and melanoma, respectively. Normal cell lines can be used as controls for cell-based studies if necessary. HEK293 can be utilized for “normal” kidney cells. They are not quite “normal” as they are transfected with SV40 to be immortal and form tumors under the skin in mice. But in vitro this is an acceptable model. Of course, there is nothing to prevent cytotoxicity testing in any number of additional lines, although this does not seem necessary in the initial stages of evaluating the compound’s usefulness (including lack of toxicity) as a potential boron carrier. Finally, an appropriate assay format for in vitro toxicity assessments remains to be selected. From our experience, the semi-automatic method of determining cell viability with the xCELLigence RTCA system (Agilent Technologies, Santa Clara, CA, USA), which enables continuous, quantitative, real-time cell analysis, seems to be particularly useful.

Among the colorimetric cell viability tests, the most popular is the 3-(4,5-dimethylthiazol-2-yl)-2,5-diphenyltetrazolium bromide (MTT) assay, which is based on mitochondrial oxidoreductase activity and the reduction of yellow MTT to purple formazan in living cells.

Therefore, in this guide, we propose a neutral red uptake test that is independent of metabolism, based on the incorporation of the neutral red dye into lysosomes, and still the gold standard for assessing cell viability/cytotoxicity. It is based on the ability of viable cells to incorporate and bind the neutral red dye in lysosomes. Most primary cells and cell lines from diverse origins may be used successfully. The procedure is cheaper and more sensitive than other cytotoxicity tests (tetrazolium salts, enzyme leakage, or protein content). Once the cells have been treated, the assay can be completed in less than 3 h. An example procedure for testing cellular toxicity using a neutral red assay is shown in the experimental section below.

#### 2.1.6. Cellular Uptake

One of the required and indispensable requirements of a promising boron carrier for BNCT is its ability to achieve therapeutic concentrations that are on average estimated at ~10^9 10^B atoms per cancer cell. It is important to remember, however, that cancer is a dynamic disease and that cancers generally become more heterogeneous as they progress. As a result of this heterogeneity, the main tumor may contain a large number of cells that have different molecular signatures and react differently to treatment, partly due to the different uptake of drug molecules by the individual cancer cells [31].

At the initial stages of testing boron compounds, the ability to penetrate cell membranes is estimated based on in vitro tests in various cell lines. In this context, it is worth remembering that heterogeneity, which is sometimes significant, also applies to cells in cell cultures. The boron content in cells is most often determined using one of the variants of mass or atomic spectrometry techniques, such as inductively coupled plasma mass spectrometry (ICP-MS), inductively coupled plasma atomic emission spectrometry (ICP-AES), direct current plasma AES, flow-injection electrospray tandem mass spectrometry, and secondary ion mass spectrometry. The use of disposable borosilicate glass free or plasticware is required to prevent boron cross contamination in the sample [32]. Each of these methods has its characteristics and scope of applications, e.g., ICP-AES is sufficient for standard boron content measurements, while ICP-MS can discriminate between ^10^B and ^11^B, though this is usually not necessary in the initial stages of boron carrier evaluation. It’s worth remembering that ICP-AES provides a higher detection limit down to ppm or ppb, whereas ICP-MS provides a lower detection limit down to ppt. Most recently, considering the heterogeneity of cells in cell cultures and their different abilities to uptake boron compounds, single-cell ICP-MS (SC-ICP-MS) was used to assess the boron content in individual cells of the same culture.

To initially estimate the penetration of boron carriers into cells and facilitate the comparison of their ability to penetrate the cell membrane, we propose the use U87MG and HEK293 cells as well as standard ICP-AES. If necessary, these studies can be supplemented with the use of dedicated cell lines, if a given compound was designed for a specific biological target, other ICP techniques, or other methods for boron concentration measurements such as PGA, or the use of specific molecular probes [15]. Subcellular localization is another important criterion because DNA damage by alpha rays generated from the boron neutron capture reaction strongly affects cell killing although standard evaluation protocols have not yet been established [15]. An example ICP-AES protocol for assessing cellular uptake is shown in the experimental section below.

### 2.2. Protocols Suggested

#### 2.2.1. Solubility Determinations

The stock solutions in DMSO (10 mM) of the selected drug candidates were diluted to decreased molarity, from 300 to 0.1 μM, in a 384-well transparent plate (Greiner 781801) with 1% DMSO: 99% PBS buffer. These were incubated at 37 °C and read after 2 h. The results were adjusted to a segmented regression to obtain the maximum concentration in which compounds are soluble. Digossin, prazosin, and progesterone were used as reference compounds (equilibrium solubilities = 84.0, 62.8 and 6.5 mM, respectively). A detailed description of the proposed nephelometric method is available in the work by Dehring et al., among others [33].

#### 2.2.2. pK_a_ Determination

pK_a_ measurements were performed on the Pion SiriusT3 system (Pion Inc. Ltd., Forest Row, UK) using the spectrometric technique [22,24,34]. Acidity constants (pK_a_ values) were determined by titration of the fully dissolved drug for UV-active ionizable groups between pH 2 and 12 at a 2 mM concentration of the compound, which was achieved by adding 5 μL of 10 mM stock solution to 25 μL of Neutral Linear Buffer. Spectrometric pK_a_ values were obtained from UV absorption measurements as a pH function applying the target factor analysis methodology [24]. Potentiometric pK_a_ values were derived from titration curves by applying charge and mass balance equations, and the pK_a_ value that provided the best fit of calculated titration data to the measured ones was taken as the final pK_a_ value. The pK_as_ value corresponds to the average pK_a_ from a minimum of three individual results. All measurements were taken at 25 °C, under an inert gas atmosphere of argon, and at least three titrations were made for each compound. A detailed description of the spectrometric method of pK_a_ determination is available in the Sirius T3 manual [25] and in Isik et al. [35].

#### 2.2.3. Log P/D Determination

An aqueous pH 7.4 phosphate-buffered solution was prepared and then saturated with n-octanol. Likewise, an n-octanol solution saturated with pH 7.4 aqueous phosphate buffer was also prepared. All the compounds were prepared as 10 mM solutions in DMSO, which was taken as a stock drug solution. The compound stock solution was diluted in the aqueous pH 7.4 phosphate buffer at a 1:100 volume ratio. This solution was taken as a standard solution. From it, different partitions were made with different *n*-octanol/water ratios according to the approximate log D_7.4_ value of the drug. Partitions were shaken for one hour at 25 °C with a rotation shaker in chromatographic vials (1.5 mL). Both the standard solution (conveniently diluted, if necessary) and the aqueous phase of each partition after equilibration were HPLC chromatographed for analysis. A detailed description of the proposed methodology is available in Andres et al. [21].

#### 2.2.4. Stability of the Compounds in Human Plasma

This protocol was derived from a previously published procedure by Patra et al. [26] and Keller et al. [27]. The stabilities of the compounds were evaluated with caffeine as an internal standard. The pooled human plasma was obtained from Bio-west and caffeine from TCI Chemicals. For each experiment, fresh stock solutions were prepared in DMSO and water, for the compound and caffeine, respectively. Following this, 25 μL of the solution containing the studied compound (5.0 mM) and 25 μL of the caffeine solution (5.0 mM) were added to 950 μL of plasma to reach a total volume of 1000 μL. The resulting solutions were incubated for 0 h, 1 h, 3 h, 6 h, 18 h, and 24 h at 37 °C with continuous and gentle shaking (ca. 500 rpm) while protected from light. Subsequently, the plasma solution was quenched with 1 mL MeOH and 2 mL CH_2_Cl_2_, and the mixture was shaken for 15 min at 25 °C followed by centrifugation at 6000 rpm for 10 min. The organic layer was separated from the aqueous layer and the CH_2_Cl_2_ was removed under reduced pressure. The obtained residue was dissolved in 200 μL CH_3_CN. The solution was filtered through a 0.2 μm membrane filter and analyzed using a 1260 Infinity HPLC System (Agilent Technology, Santa Clara, CA, USA). A Pursuit XRs C18 (250 × 4.6 mm) reverse phase column was used at a flow rate of 1 mL/min [27].

#### 2.2.5. Cellular Toxicity by Using the Neutral Red Assay

As an example, the protocol for cytotoxicity assignment in HEK293 cells is described. Following disaggregation of cells with trypsin/EDTA and resuspension of cells in the medium, a total of 9 × 10^3^ cells/well were plated in 96 well tissue-culture plates and the plates were placed into a 37 °C, 5% CO_2_ incubator. After 24 h incubation, the different concentrations of tested compounds in the medium were added. The cells were incubated for 48 h at 37 °C in 5% CO_2_, then the medium was aspirated. and cells were washed twice with PBS, and incubated for 3 h in a medium supplemented with neutral red (50 mg/mL). Next the medium was washed off rapidly with a solution containing 0.5% form aldehyde and 1% calcium chloride. Cells were subjected to further incubation of 20 min at 37 °C in a mixture of acetic acid (1%) and ethanol (50%) to extract the dye. The absorbance of the solution in each well was measured in a microplate reader at 540 nm and compared with the wells containing untreated cells. Results were expressed as the mean percentage of cell growth inhibition from three independent experiments. Cell viability was plotted as the percent of control (assuming data obtained from the absence of compounds as 100%). More detailed, manual-type procedures may be found in Al-Sheddi et al. [36], Repetto et al. [37], or Ates et al. [38].

#### 2.2.6. Boron Cellular Uptake Using the ICP-AES Method

The mouse colorectal carcinoma cell line, colon 26, was maintained at 37 °C under 5% CO_2_ atmosphere in RPMI 1640 medium supplemented with 10% fetal bovine serum FBS, HyClone,), 100 U/mL of penicillin, and 100 mg/mL of streptomycin (Invitrogen, Carlsbad, CA, USA). For subsequent experiments, the cells were seeded at a density of 1 × 10^5^ cells in a 60 mm diameter dish (Greiner, Kremsmünster, Austria) and incubated at 37 °C for 24 h. Three dishes were used for cell counting (1.6 × 10^6^ cells/dish). The cells in the other dishes were incubated at 37 °C in the presence of various boron concentrations of the boronated DMPC-liposomes for another 1 h and then washed three times with phosphate-buffered solution (PBS). The cells were digested with 2 mL of perchloric acid/hydrogen peroxide at 70 °C for 6 h and then the digested samples were diluted with distilled water. After filtering through a hydrophobic filter (13JP050AN, ADVANTEC, Tokyo, Japan), the boron concentration was measured by using ICP-AES. The BSH-encapsulated DMPC-liposomes (BSH-liposome) prepared according to the literature procedure were used as a control experiment [39]. A detailed SC-ICP-MS procedure for single cell boron contents measurements may be found in Balcer et al. [40].

**Note:** The important thing about using ICP methods for boron concentration analyses in biological samples is the nature of the boron compound used. Though single boron atom-containing compounds, e.g., derivatives of boronic acids, are readily degraded for ICP-MS analysis in strong acid solution, but boron clusters such as carboranes are difficult to degrade, resulting in inaccurate measurement of boron content. In such cases, it is recommended to use much stronger degradation conditions such as suspension of the sample in HNO_3_ (70%) and H_3_PO_4_ (85%) and subjection to several high-pressure microwave digestion cycles at 200 °C and 240 °C. The detailed procedure is described in Morrison et al. [41].

## 3. Discussion

For decades, BNCT could only be carried out in research reactors because only these could provide the quality and quantity of neutrons required for BNCT. With advances in accelerator technology, this has fundamentally changed and now neutron sources for BNCT can be set up in hospitals. This and the attractive principle of BNCT, namely the selective destruction of cancer cells while sparing adjacent normal cells, make this method appealing to the medical community, rapidly increasing the demand for such neutron sources for BNCT worldwide. This creates a market for boron carriers, which has not existed up to now. So far, only two drugs have been available for the treatment of patients [8], sodium mercaptoundecahydro-*closo*-dodecaborate (BSH, Na_2_^10^B_12_H_11_SH), which was synthesized by Soloway in 1967 [42] and (^10^B)-4-Borono-l-phenylalanine (BPA, C_9_H_12_^10^BNO_4_), which was synthesized by Snyder et al. and published in 1958 [43]. Neither of them were available on the market as a drug. It was not until 2020 that Stella Pharma Corporation launched, Steboronine^®^ with generic name borofalan(^10^B), which contains highly boron-10 enriched boronophenylalanine (^10^BPA) as the active pharmaceutical ingredient (API). However, with a market for boron carriers appearing at the horizon, the interest in synthesizing new boron compounds for BNCT is growing. Most of this research is conducted at academic institutes and not by industry. However, there, only initial tests can be carried out to decide whether a new substance is a promising drug candidate and if it should be included in the complex and expensive further test phases for a new drug. Until now, these early tests were carried out in such a way that the results could not be directly compared, which is a prerequisite for selecting the most promising candidates for further development. The guideline we propose covers this very early phase of a drug development but includes first experiments with cells, which is a crucial step for further efforts. The cell lines we suggest for this early testing were selected to cover the current most promising indications for BNCT, human malignant glioma cells U87MG as an example of a high-grade glioma, which still is perceived as target indication for BNCT. Furthermore, squamous cell carcinoma, SAS, as an example of head and neck squamous carcinoma, and finally A375 for melanoma, which seem to be attractive indications for BNCT. HEK293 cell line might be used as an example of normal tissue cells.

The list of proposed tests can certainly be expanded, although this does not seem strictly necessary at the initial stage of boron drug evaluation. Also, the choice of specific methods for measuring specific physicochemical properties may be subject to discussion. It would be best if identical methods of measuring these parameters were always used everywhere, though this is not in every case possible and practical. An example would be the measurement of solubility. Conventional thermodynamic solubility measurements based on achieving equilibrium between a solid and a liquid medium are not feasible in the early phase of discovery because large sample volumes are required, throughput is low, and sample preparation is labor intensive. An alternative to equilibrium solubility is the kinetic solubility proposed here, where compounds are pre-dissolved in dimethyl sulfoxide (DMSO) and the solubility is measured as the concentration at which the sample precipitates from an aqueous medium. The kinetic solubility measurement is not intended to serve as a substitute for a thermodynamic solubility value because crystal lattice effects are negated when the compound is dissolved in DMSO, but it is acceptable at the early stages of the drug screening.

Therefore, we leave the guide proposal at this stage of detail, being aware of its imperfections but hoping that it will be useful both for investigators who have been working on boron carriers for a long time and for those who join the developing BNCT field in the near future.

## 4. List of Example Laboratories Where Preliminary Bioprofiling Tests of Potential Boron Carriers Can Be Carried Out

Boron agents are synthesized in chemistry laboratories, and not all laboratories are equipped to perform the proposed tests. To facilitate the early profiling procedure we include a sample list of academic laboratories where such tests can be performed and which can give some support.

### 4.1. Solubility in H_2_O

Faculty of Pharmacy ULisboa, Portugal, https://imed.ulisboa.pt/members/rui-moreira/ (accessed on 4 March 2024), kinetic solubility, HPLC, Shimadzu LC-2050 (Shimadzu Europa GmbH, Duisburg, Germany).INNOpharma Platform for Drug Screening, University of Santiago de Compostela, Spain, ES-Openscreen, https://www.usc.es/biofarma/ (accessed on 4 March 2024), nephelometric method.Latvian Institute of Organic Synthesis, Latvia, https://www.osi.lv/en/services/analytical-chemistry/ (accessed on 4 March 2024), both kinetic and thermodynamic solubility assessment by direct concentration measurement in selected buffer solutions by HPLC/UV, Waters Alliance Separation module (Waters Co., Milford, MA, USA).LC-MS Metabolomics Center, School of Pharmacy, University of Eastern Finland, Finland, https://uefconnect.uef.fi/en/group/lc-ms-metabolomics-center/#information, (accessed on 4 March 2024), equilibrium shake flask thermodynamic solubility method, HPLC-UV/LC-MS (Agilent QQQ 6495) methods.National Library of Chemical Compounds POL-OPENSCREEN, Institute of Medicinal Biology PAS, Poland, https://pol-openscreen.pl (accessed on 4 March 2024), nephelometric method.

### 4.2. pK_a_ Determination

INNOpharma Platform for Drug Screening, University of Santiago de Compostela, SpainES-Openscreen, https://www.usc.es/biofarma/ (accessed on 4 March 2024), simultaneous determination of the UV spectra as a function of pH, EnVision Multilabel Reader.Latvian Institute of Organic Synthesis, Latvia, https://www.osi.lv/en/research/research-areas/physical-organic-chemistry/ (accessed on 4 March 2024), Nuclear Magnetic Resonance (NMR), 400 MHz or 600 MHz NMR system equipped with cryoprobes for better sensitivity (Bruker, Billerica, MA, USA).National Library of Chemical Compounds POL-OPENSCREEN, Institute of Medicinal Biology PAS, Poland, Poland, https://pol-openscreen.pl (accessed on 4 March 2024), potentiometric and spectrophotometric method, Pion SiriusT3 (Pion Inc. Ltd., Forest Row, UK).

### 4.3. Log P/D Determination

Faculty of Pharmacy ULisboa, Portugal, https://imed.ulisboa.pt/members/rui-moreira/ (accessed on 4 March 2024), shake flask method, HPLC, Shimadzu LC-2050 (Shimadzu Europa GmbH, Duisburg, Germany).Latvian Institute of Organic Synthesis, Latvia, https://www.osi.lv/en/services/analytical-chemistry/ (accessed on 4 March 2024), shake flask method with HPLC/UV concentration measurements, Waters Alliance separation module (Waters, Milford, MA, USA).LC-MS Metabolomics Center, School of Pharmacy, University of Eastern Finland, Finland (https://uefconnect.uef.fi/en/group/lc-ms-metabolomics-center/#information (accessed on 4 March 2024), shake flask method for lipophilicity, HPLC-UV/LC-MS (Agilent QQQ 6495) methods.National Library of Chemical Compounds POL-OPENSCREEN, Institute of Medicinal Biology PAS, Poland, https://pol-openscreen.pl (accessed on 4 March 2024), potentiometric titration method, Pion SiriusT3 (Pion Inc. Ltd., Forest Row, UK).POL-OPENSCREEN, Institute of Biochemistry and Biophysics PAS, Poland, https://pol-openscreen.pl (accessed on 4 March 2024), hydrophobicity, reverse phase HPLC analysis (Knauer, Berlin, Germany).

### 4.4. Stability of Compounds in Buffers and Human Plasma

Faculty of Pharmacy ULisboa, Portugal, https://imed.ulisboa.pt/facilities/ (accessed on 4 March 2024), HPLC and LC-MS, triple Quadrupole Micromass Quattro Micro API (Waters, Milford, MA, USA).INNOpharma Platform for Drug Screening, University of Santiago de Compostela, Spain (ES-Openscreen, https://www.usc.es/biofarma/ (accessed on 4 March 2024)), UPLC-MSMS/DAD.Latvian Institute of Organic Synthesis, Latvia, https://www.osi.lv/en/services/analytical-chemistry/ (accessed on 4 March 2024), LC/MS/MS, Waters Xevo-TQS (Waters, Milford, MA, USA).LC-MS Metabolomics Center, School of Pharmacy, University of Eastern Finland, Finland (https://uefconnect.uef.fi/en/group/lc-ms-metabolomics-center/#information (accessed on 4 March 2024)), HPLC-UV/LC-MS (Agilent QQQ 6495) methods.National Library of Chemical Compounds POL-OPENSCREEN, Institute of Medicinal Biology PAS, Poland, https://pol-openscreen.pl (accessed on 4 March 2024), LC-MS method, Agilent 6546 LC/Q-TOF (Santa Clara, CA, USA).POL-OPENSCREEN, Institute of Biochemistry and Biophysics PAS, Poland, https://pol-openscreen.pl (accessed on 4 March 2024), stability in buffers of various pH, LC/MS/MS quantitative method, mass spectrometer: Xevo TQ-S with standard EIS ion source (Waters), chromatograph: Acquity M-class (Waters).

### 4.5. Cellular Toxicity

Faculty of Pharmacy ULisboa, Portugal, https://imed.ulisboa.pt/facilities/ (accessed on 4 March 2024), cell viability MTT assay, GloMax^®^-Multi+Microplate Reader (Promega Co., Madison, WI, USA).INNOpharma Platform for Drug Screening, University of Santiago de Compostela, Spain (ES-Openscreen, https://www.usc.es/biofarma/ (accessed on 4 March 2024)), cellular cytotoxicity measured using one healthy, and two cancer cell lines. Cell viability assays: MTS, MTT, CellTiter Glo.Screening Laboratory POL-OPENSCREEN, Institute of Medicinal Biology PAS, Poland, https://pol-openscreen.pl (accessed on 4 March 2024), xCELLigence RTCA system (Agilent Technologies, Santa Clara, CA, USA), real-time cell viability method or neutral red method.

### 4.6. Boron Cellular Uptake

Biological and Chemical Research Centre, University of Warsaw, Poland, ICP-MS method, NexION 300D, PerkinElmer (Waltham, MA, USA).Laboratory of Chemistry of the Institute of Food Safety, Animal Health and Environment “BIOR”, Latvia, https://bior.lv/en (accessed on 4 March 2024), ICP-MS method, Thermo Scientific ICAP™ RQ ICP-MS (Waltham, MA, USA) and Agilent 7700 ×ICP-MS ((Santa Clara, CA, USA).LC-MS Metabolomics Center, School of Pharmacy, University of Eastern Finland, Finland (https://uefconnect.uef.fi/en/group/lc-ms-metabolomics-center/#information (accessed on 4 March 2024)), ICP-MS method, NeXION 350D (PerkinElmer Inc., Waltham, MA, USA).Dipartimento di Scienze e Innovazione Tecnologica, Università del Piemonte Orientale, Alessandria, Italy, ICP-MS method, Thermo Scientific iCAP RQ ICP_MS (Waltham, MA, USA); ICP-OES method: Spectro Genesis (AMETEK, Berwyn, PA, USA).Graduate School of Interdisciplinary Science and Engineering in Health Systems, Okayama University, Okayama, Japan. Contact info: eijimatu.01@gmail.com, ICP-AES method.Open Facility Center, Tokyo Institute of Technology, Japan, https://www.ofc.titech.ac.jp/en/ (accessed on 4 March 2024), ICP-MS method, ICP-MS ELAN-DRC-es (PerkinElmer Inc., Waltham, MA, USA).

## Figures and Tables

**Figure 1 cells-13-00798-f001:**
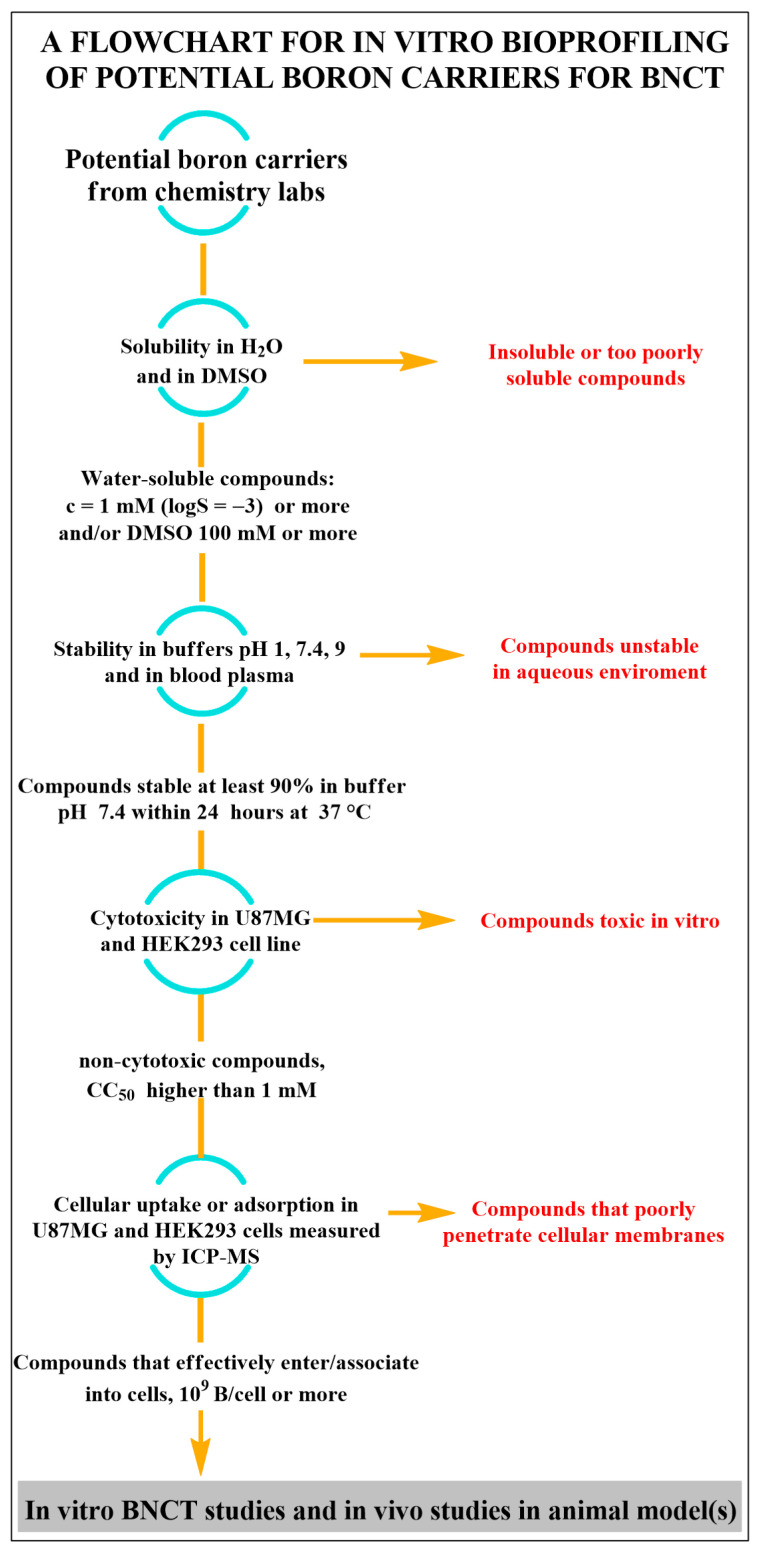
Flow chart for in vitro bioprofiling of potential boron carriers for BNCT.

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
