# Peer review of "Early Stage In Vitro Bioprofiling of Potential Low-Molecular-Weight Organoboron Compounds for Boron Neutron Capture Therapy (BNCT)—Proposal for a Guide"

_cells, 2024, doi:10.3390/cells13100798_

Round 1

Reviewer 1 Report

Comments and Suggestions for Authors

In this manuscript, the authors propose to standardize an early evaluation method for the different boron carriers currently being developed by many laboratories. This fact makes it necessary to standardize the methodology for testing these new compounds, as each laboratory may use different evaluation methods and protocols. This lack of standardization can hinder the comparison of the properties of different boron compounds and therefore make it difficult to make a correct decision when selecting the most suitable ones.

The authors' proposal is very interesting, but while evaluating the manuscript, I have encountered some doubts that the authors need to address.

The steps proposed by the authors seem correct to me, although I have found that the explanation of some of these steps and the procedure to follow need a deeper explanation.

- Authors should explain why they suggest a concentration of 1mM for dissolution in water and 100mM for dissolution in DMSO. I believe that the authors are generalizing too much and should provide examples of compounds to guide the reader, as each boron compound is different.

-Why do the authors propose only water and DMSO? Boron compounds present much difficulty in being soluble in these two solvents. The authors should explain why they simplify this section.

-Sections 2.1.2 and 2.1.3 seem very interesting and valid for the evaluation of these compounds. It would be very interesting if the authors explained the optimal values of these measurements or if they depend on the type of tissue or cell line and the compound itself.

-Authors should explain why they consider a CC50 value of at least 100 µM as a criterion for cytotoxicity for a compound classified as non-toxic. Why that concentration? Can it be generalized?

-Lines 257-260. The authors claim that it is not necessary to test the compound's utility as a possible boron carrier in different cell lines in the initial stages. They propose certain lines. The fact that there is a difference in the response to boron compounds in other studies in different cell lines necessitates checking the compound so that unexpected results are not encountered when the research is more advanced.

-The authors propose two different methods for studying cell viability. I advise the authors to propose a method in which toxicity is determined by measuring cell apoptosis, for example, with ANEXINAV/PI using flow cytometry. The results are very reliable, and the protocol is very fast.

Author Response

Response to Reviewer 1

Thank you for your kind comments and suggestions, which have enabled us to identify some points that require clarification. You will find our detailed responses below

Reviewer 1

- Authors should explain why they suggest a concentration of 1mM for dissolution in water and 100mM for dissolution in DMSO. I believe that the authors are generalizing too much and should provide examples of compounds to guide the reader, as each boron compound is different.

Thank you for this remark.  Indeed, our goal was to propose a very general set of rules for early-stage, bioprofiling of boron compounds. Evaluation at this stage is of a screening nature, a more detailed (less general) study is justified after a preliminary selection of compounds that show to a greater or lesser extent, the basic required for boron carrier properties.

The proposal to select the minimum solubility in water of compounds accepted for further testing at the level of 1 mM is based on a simple criterion, physicochemical and in vitro tests of biological properties are carried out in aqueous solutions - compounds insoluble in water cannot be tested. Also at further stages of drug development in vivo, good solubility in water is desirable, though low solubility can be overcome to some extent by an appropriate drug formulation.

The solubility (including solubility in water)  in most pharmacopeias is measured by the ratio of the mass of the compound to the mass of the solvent (msv/msu) and ranges from Very soluble <1 to Practically insoluble or insoluble ≥ 10,000. We chose Slightly soluble range 100 to 1000 as the limit value which corresponds to 1–10 mg /mL, and assumed the upper limit of this range 10 mg/mL.

Since in chemistry molar concentrations are used more often than percent concentration, the concentration of 10 mg/mL was converted into a molar concentration for an exemplary compound with a molecular weight of 500 Da, which is the conventional upper limit for the optimal mass of chemotherapeutics according to Lipinski's rule. 1 mM is a low solubility but still allows to work with the compound. The point is not to reject too many compounds at the start.

Concerning the remark that authors “should provide examples of compounds to guide the reader, as each boron compound is different”,  we completely agree with the statement that “each boron compound is different”, therefore not every compound containing boron is a good candidate as a boron carrier for BNCT.

In the solubility in water part, the statement was added: The solubility in water value of 1 mM is within the range defined as “slightly soluble”. This solubility, although low, allows the testing of boron compounds and does not result in the rejection of too many derivatives at the beginning of the boron compound evaluation. For comparison, the solubility of BPA which is considered highly insoluble in water, is 2.8 – 3.3 mM (0.6-0.7 mg/mL [9].

-Why do the authors propose only water and DMSO? Boron compounds present much difficulty in being soluble in these two solvents. The authors should explain why they simplify this section.

The choice of water and DMSO as a solvent in which the solubility of boron compounds should be determined primarily is explained in the first two sentences of the "Solubility in water" section: “Solubility in water, the main component of biological fluids, is a key physicochemical property for the success of any drug candidate. It determines the compound absorption, bioavailability, and safety. The assessment of solubility in dimethyl sulfoxide (DMSO), in turn, is needed because it is a standard solvent used to prepare stock solutions that are then used in most physicochemical and biological in vitro assays.”

Most in vitro tests are performed in an aqueous environment and stock solutions are usually prepared in DMSO as the solvent. Compounds that are insoluble in water cannot be tested under routine assay conditions. Other solvents, except ethyl alcohol which is occasionally used, are not useful in the preparation of stock solutions for biological assays.

-Sections 2.1.2 and 2.1.3 seem very interesting and valid for the evaluation of these compounds. It would be very interesting if the authors explained the optimal values of these measurements or if they depend on the type of tissue or cell line and the compound itself.

pKa of a drug is a key physicochemical parameter influencing many bio-pharmaceutical characteristics. Answering the reviewer's simple question is not easy. The optimal pKa value may depend on the route of drug administration and the target site in the patient's body, among others. Within the WHO essential medicines list 77.5 % of drugs had an ionizable group with a pKa in the range 2–12 which covers almost the entire range of possible pKas.

The same is in the case of lipophilicity (logP), according to Lipinski's rule of five, the lipophilicity of drugs like compounds should be within the log P −0.4 to +5.6 range. The optimal lipophilicity depends on the specific case.

-Authors should explain why they consider a CC50 value of at least 100 µM as a criterion for cytotoxicity for a compound classified as non-toxic. Why that concentration? Can it be generalized?

Classifying a compound as cytotoxic or not often depends on the biological context and the compound’s specific future application. One of the classifications of toxicity covers the CC50 range: 1-20 mM, strong cytotoxicity; 20-100 mM, moderate toxicity; 100-200 mM, low toxicity; above 200 mM, non-toxic (Profiles of Drug Substances, Excipients and Related Methodology, 2021, 46, pp. 273-307). However, the interpretation of CC50 cytotoxicity parameter also depends on whether we are looking for an antiviral, antibacterial, anticancer molecule or boron carrier for BNCT. In the case of drugs against infectious diseases, the key parameter is the selectivity index (SI) defined as a ratio between CC50 and IC50 (the concentration of compound inhibiting viral or bacteria’s replication by 50%: SI = CC50/IC50. The high SI means that to obtain a therapeutic effect, the compound can be used in low doses, which sometimes allows the acceptance of a higher CC50 (cytotoxicity) of the compound. There is no such reduced tariff in the case of boron compounds for BNCT, these compounds should be non-toxic. The two boron carriers used so far in BNCT practice, BSH and BPA, are non-toxic according to the above-mentioned criterion, for BSH CC50 = 3.79 mM in V79 cells (Bioconjugate Chem., 2007, 18, 1287) and for BPA CC50 > 2 mM in B16 cell line (J. Med. Chem. 2012, 55, 6980).

The cellular toxicity value of 0.1 mM is on the edge of moderate toxicity range and compared to the low toxicity of BSH and BPA, it is highly non-discriminatory so as not to reject too many compounds already at the first stage of toxicity screening. The advised second stage of cytotoxicity testing in the concentration range 0.01 - 10 mM will allow to determine the CC50 and decide whether to continue the evaluation of a given compound or not.

A proposed criterion for CC50 value classified as non-toxic at 100 µM is incorrect. We changed it to 1 mM, which is still a highly tolerant value, but closer to the value range of other boron carriers. Thank you for bringing this to our attention.

In the cytotoxicity section, the statement was added: For comparison, the cytotoxicity of BSH and BPA was assessed as CC50 = 3.79 mM in V79 cells [22a] and CC50 > 2 mM in B16 cell line [22b], respectively.

-Lines 257-260. The authors claim that it is not necessary to test the compound's utility as a possible boron carrier in different cell lines in the initial stages. They propose certain lines. The fact that there is a difference in the response to boron compounds in other studies in different cell lines necessitates checking the compound so that unexpected results are not encountered when the research is more advanced.

Boron compounds are tested in different cell lines, including B16, B16F10, MKT, A1059, TA1059-1, MRA 27, SK-23 Mel, C6, U87-MG, LN-229, T98G, KB, FaDu, SAS, A-253, HuH-7, and many others. Each laboratory has a set of cell lines that best suit their needs and research profile. When we wrote, “Of course, there is nothing to prevent cytotoxicity testing in any number of additional lines, although this does not seem necessary in the initial stages of evaluating the compound's usefulness as a potential boron carrier”, we meant that there is no need to test toxicity on multiple cell lines. What is more, important is the comparability of results by using the same lines and answering the question of whether the compound is toxic or not, the prerequisite for boron carrier for BNCT. In this context, some differences between different cell lines in susceptibility to the toxic effects of compounds are of limited importance. Especially since we propose lines that are commonly used in the study of boron compounds. Further stages of compound evaluation, which include among others more advanced toxicity studies, will use their methods and protocols anyway. The purpose of the proposed guide is relatively quick and economically justified screening to select compounds for further, more detailed, and advanced (and more expensive) study.

The incriminated statement was changed to “Of course, there is nothing to prevent cytotoxicity testing in any number of additional lines, although this does not seem necessary in the initial stages of evaluating the compound's usefulness (including lack of  toxicity) as a potential boron carrier.”

-The authors propose two different methods for studying cell viability. I advise the authors to propose a method in which toxicity is determined by measuring cell apoptosis, for example, with ANEXINAV/PI using flow cytometry. The results are very reliable, and the protocol is very fast.

Thank you for the comment. In the guide, we mention two methods for testing the toxicity of compounds, one determines viability based on measuring changes in the resistance of the cell monolayer under the influence of the drug, and the other determines cytotoxicity using a standard colorimetric approach. In response to a cytotoxic substance, a cell may either cease to proliferate or die due to two main mechanisms, apoptosis or necrosis. The toxicity of a chemical agent is most often due to severe cellular damage most commonly, damage to the cell membrane. Limiting the test to one of the possible causes of cell death, in this case, apoptosis,  would not be helpful, what is important is the answer to the question of to what extent the compound causes cell death regardless of the mechanism (at the early profiling stage). The proposed simple colorimetric test provides an answer to this question. Moreover, although flow cytometers are more and more standard equipment, not all laboratories have them.

Reviewer 2 Report

Comments and Suggestions for Authors

Dear authors,

This article is devoted to the idea of creating a certain minimum list of parameters for characterizing newly developed drugs that deliver the boron isotope during BNCT. Indeed, when developing a particular drug, one would like to unify the characterization procedure and compare other boron delivery agents developed in other laboratories around the world using the same parameters. Something like this is published all the time, for example, regarding methods - doi: 10.1373/clinchem.2008.112797, https://doi.org/10.1371/journal.pbio.3001783https://doi.org/10.1038/s41596-022-00700-y. You can then create a database of such drugs and conduct a real meta-analysis. Of course, this is an opportunity for greater cooperation between research groups and a good basis for greater openness of research. There is already an excellent article written in this direction -https://doi.org/10.1093/jrr/rrad064.

That is, the preamble of the article is wonderful, but there are some substantive comments that allow me to limit myself to the status of minor revision or accepted:

1. L. 27-28. I think we need to add aggregation, particle size and charge to this list, especially for nanoparticles, liposomes or nanotubes. Characterization can be done using dynamic light scattering as well as electron microscopy. It is better to use the word hydrophobicity rather than lipophilicity; again, it must be clarified if the drug being developed must pass the BBB. I don’t see any point in specifying the dimensions of the boron deliverers in the abstract.

2. L. 78 -82. Taking into account the fact that the dispersion of particles after neutron capture is 10 microns, that is, approximately equal to the diameter of the cell, it is strange to write that «Targeted boron agents (DNA, mitochondria) likely will require lower levels than this». If there is such a quantity inside, then they will also enter the mitochondria and the nucleus. But to prove the opposite will be quite labor-intensive - it is necessary to separate the nuclear and mitochondrial fractions separately.

3. L. 93-95. You can also look at new reviews and expand the palette of promising compounds presented in this paragraph. Here you do not indicate all that are promising for further research. (https://doi.org/10.3390/cancers15133277 etc.).

4. 2.1.1. Solubility in H2O. In this section, clarify that you mean the molar solubility of a substance in water mole/liter. Solubility is also expressed in gm/liter or g/dm3 or g/100g H2O. It would also be great to clarify the solubility of substances in DMSO. In the same subsection, it is necessary to indicate restrictions on the use of DMSO, for example, CC50 for cell lines («The IC50 of DMSO (concentration which produces 50% inhibition of growth) was 1% for smooth muscle cells and 2.9% for endothelial cells». DOI: 10.1007/BF02623858). DMSO is toxic to cells, and in pursuit of solubility, we can kill cells with the solvent (DOI: 10.1016/j.toxrep.2018.10.002).

 5. Figure 1 should be included in the introduction, but not in the section on solubility.

 6. It is better to indicate the method, its physical basis, rather than advertise the manufacturer - NEPHELOstar Plus (BMG LABTECH) (https://doi.org/10.5599/admet.4.2.292, DOI:10.1021/acs.jced.8b01263, https://courseware.cutm.ac.in/wp-content/uploads/2022/01/Importance-of-Solubility.pdfhttps://doi.org/10.1016/j.drudis.2022.01.017).

 7. Regarding pKa, the same thing, indicate conditions, parameters, ranges and refer to generally accepted measurement standards, methods (doi: 10.4137/ACI.S12304). So far it feels like you are advertising the Pion SiriusT3 system (Pion Inc. Ltd., 324 Forest Row, UK).

8. Regarding Determining Partition Coefficient (Log P), Distribution Coefficient (Log D) and Ionization Constant (pKa) in Early Drug Discovery, of course you can recommend HPLC, but not every laboratory has it, it is an expensive method. For exploratory or proof-of-principle work, you can do without this. It is clear that when a drug is at the preclinical testing stage, it will be necessary to fully characterize the drug, including HPLC.

 9. 2.1.4. Evaluation of the compound's stability at various pH and in human plasma, 2.2.4. Stability of the compounds in human plasma. The same thing - there is a classic, widespread, cheaper method for determining nuclease activity using electrophoretic analysis.

 10. 2.1.5. Cellular toxicity. First, please explain why these particular cell lines - for example, T98G, and not U87MG, not U343, not U251, etc. (https://www.atcc.org/search#q=glioblastoma&sort=relevancy&numberOfResults=24). Why such a sarcoma (https://www.atcc.org/search#q=squamous%20cell%20line&sort=relevancy&numberOfResults=24). Why this particular control? (https://www.atcc.org/search#q=Vero&sort=relevancy&numberOfResults=24). These cells are of mesenchymal origin, but the main thing is that it is obtained from the epithelium of the kidney taken from the African green monkey (Chlorocebus aethiops), that is, not a person! Secondly, refer to generally accepted methods - (Assay Guidance Manual. Markossian S, Grossman A, Brimacombe K, et al., editors. Bethesda (MD): Eli Lilly & Company and the National Center for Advancing Translational Sciences; 2004), which does not describe the use of neutral red. Judging by your protocol, you recommend selecting apoptotic cells detached from the plastic, leaving only living ones attached to the surface of the plastic, and staining the lysosomes? Which is not entirely correct, since you are taking some of the cells, usually a tetrazolium salt preparation is dripped on top, without affecting the cells in the wells of the plate. Please provide a link to a study that shows the absence of interaction of eurhodin dye with boron clusters, similar to how you write about MTT (“However, the redox activity of some boron carriers, especially those 267 with a boron cluster in their structure, may overlap with the process of MTT reduction to 268 formazan, thus distorting the readings of cell viability").

 11. Regarding 2.1.6. Cellular uptake, then both methods (ICP-MS, ICP-AES) are suitable, the main thing is to indicate the sensitivity of the device, but for some reason at some point you lost the mention of ICP-MS (L.128). Add please.

12. For some reason, your pipeline does not include an assessment of the cytotoxicity of BNCT against cells using the developed drugs - a clonogenic test (doi: 10.1038/nprot.2006.339) or RTCA xCelligence. It is worth noting that here, after irradiation, the MTT test is just not suitable (DOI:10.1038/s41598-018-19930-w)!

 13. Further, after reading the last section, I was perplexed - is this a commercially commissioned article, an advertisement? Why can’t these tests be done anywhere else? After reading your list, how would researchers feel who have three ICP-AES within walking distance but are not listed on your list? Another point, maybe this is a list of laboratories affiliated with the Ministry of Health for performing preclinical tests and they are all done according to GFP standards? Explain what their preference is? They should be removed from the article altogether, since this is advertising!

Author Response

Thank you for your kind comments and suggestions, which have enabled us to identify some points that require clarification. We are also very grateful for the discussion you have initiated.

You will find our detailed responses below

Reviewer 1

  1. L. 27-28. I think we need to add aggregation, particle size and charge to this list, especially for nanoparticles, liposomes or nanotubes. Characterization can be done using dynamic light scattering as well as electron microscopy. It is better to use the word hydrophobicity rather than lipophilicity; again, it must be clarified if the drug being developed must pass the BBB. I don’t see any point in specifying the dimensions of the boron deliverers in the abstract.

We agree, that for nanoparticle-type boron carriers (including liposomes and nanotubes) parameters such as aggregation, particle size, and charge (zeta potential) should be measured. However, this type of boron carrier is governed by different rules than low-molecular-weight organoboron compounds. For this reason, we stressed in the abstract that ".... the specific methods have been selected primarily for low molecular weight boron carriers; in the case of high molecular weight compounds, some of the methods may need to be adapted." In our opinion, the nanoparticle and polymeric boron carriers deserve a separate guide.

Concerning hydrophobicity vs. lipophilicity, both terms are often used interchangeably, although they are not the same thing. Hydrophobicity is a component of lipophilicity. In practice, when one determines logP by measuring the partition coefficient, one measures lipophilicity, so we would like to keep this term.

As to the requirement for penetration BBB, brain tumors are one of the biological targets of BNCT, albeit, requiring the ability to pass the BBB does not seem necessary at the early stage of boron compounds evaluation. The purpose of this preliminary testing is only to determine whether the boron compound meets the necessary minimum of required properties to be considered as a "potential boron carrier" at all.

  1. L. 78 -82. Taking into account the fact that the dispersion of particles after neutron capture is 10 microns, that is, approximately equal to the diameter of the cell, it is strange to write that «Targeted boron agents (DNA, mitochondria) likely will require lower levels than this». If there is such a quantity inside, then they will also enter the mitochondria and the nucleus. But to prove the opposite will be quite labor-intensive - it is necessary to separate the nuclear and mitochondrial fractions separately.

With random distribution within the cell, it is estimated that about 109 atoms of 10B are required to kill tumor cells. However, boron carrier molecules are not distributed evenly in the cytoplasm. Inside the cell, there are many compartments and organelles surrounded by membranes with specific properties. Crossing the outer cell membrane does not mean that the compound will enter the nucleus, mitochondria, endoplasmic reticulum etc. From the point of view of the BNCT effectiveness the most interesting is the nucleus containing the genetic material of the cell. Boron delivery agents that specifically target DNA are expected to significantly reduce the intracellular amount of 10B needed (Curr. Oncol. 2022, 29, 7868; J.  Nanobiotechnology, 2022, 20,102). Imaging of the intracellular boron distribution is not easy, helpful at least may be techniques such as dynamic SIMS ion microscopy or Raman microscopy (for boron cluster compounds).

  1. L. 93-95. You can also look at new reviews and expand the palette of promising compounds presented in this paragraph. Here you do not indicate all that are promising for further research. (https://doi.org/10.3390/cancers15133277 etc.).

Thank you for drawing our attention to the excellent review published recently (Cancers 2023, 15, 3277), of course, we know it. However, the purpose of the proposed guide is not to review the current status of boron carrier development, therefore we mentioned only two that have a chance of entering pre-clinical and perhaps clinical trials.

  1. 2.1.1. Solubility in H2O. In this section, clarify that you mean the molar solubility of a substance in water mole/liter. Solubility is also expressed in gm/liter or g/dm3 or g/100g H2O. It would also be great to clarify the solubility of substances in DMSO. In the same subsection, it is necessary to indicate restrictions on the use of DMSO, for example, CC50 for cell lines («The IC50 of DMSO (concentration which produces 50% inhibition of growth) was 1% for smooth muscle cells and 2.9% for endothelial cells». DOI: 10.1007/BF02623858). DMSO is toxic to cells, and in pursuit of solubility, we can kill cells with the solvent (DOI: 10.1016/j.toxrep.2018.10.002).

The concentration type we propose is molarity (in this case, mM) as stated in the sentence “As the required minimum solubility of the compound in water, we suggest not less than 1 mM, and in DMSO a minimum of 100 mM.” To avoid misunderstanding it was changed to “As the required minimum solubility of the compound in water (molar solubility), we suggest not less than 1 mM, and in DMSO a minimum of 100 mM.”

Indeed, although DMSO is commonly used to prepare stock solutions for biological assays, it is not inert to cells. It is assumed that the concentration in the aqueous environment of a biological test should not exceed 0.5 % and preferentially was at 0.1 % level. Therefore, we proposed the minimum recommended solubility of the compound in DMSO to be 100 mM, because it allows for obtaining a maximum concentration in the first stage of cytotoxicity evaluation of 0.1 mM, while maintaining a maximum DMSO concentration of 0.1%. The second stage of the proposed cytotoxicity tests at a final concentration of the compound of 1 mM requires coming to terms with a final concentration of DMSO in the test of 1% or a much higher solubility of the boron compound in water and DMSO than the required minimum of 1 mM and 100 mM, respectively.

In the section on water solubility, the following additional information was aded: “The assessment of solubility in dimethyl sulfoxide (DMSO), in turn, is needed because it is a standard solvent used to prepare stock solutions that are then used in most physicochemical and biological in vitro assays, even if it is not inert to cells. It is therefore recommended that the final DMSO concentration in biological tests should not exceed 0.5% and preferably 0.1 %..”

  1. Figure 1 should be included in the introduction, but not in the section on solubility.

The Figure 1 was moved to the introduction section.

  1. It is better to indicate the method, its physical basis, rather than advertise the manufacturer - NEPHELOstar Plus (BMG LABTECH) (https://doi.org/10.5599/admet.4.2.292, DOI:10.1021/acs.jced.8b01263, https://courseware.cutm.ac.in/wp-content/uploads/2022/01/Importance-of-Solubility.pdf, https://doi.org/10.1016/j.drudis.2022.01.017).

We have adapted example methods from original publications and provided references to these works. Most journals require detailed information in the experimental section about the equipment used in specific measurements, including the type and name of the instrument and the name of the supplier, so we have followed this format.

  1. Regarding pKa, the same thing, indicate conditions, parameters, and ranges and refer to generally accepted measurement standards, methods (doi: 10.4137/ACI.S12304). So far it feels like you are advertising the Pion SiriusT3 system (Pion Inc. Ltd., 324 Forest Row, UK).

In the Abstract it is clearly stated that “The selection of assays and corresponding methods is based on the practical experience of the authors, who have been working for many years in various fields of BNCT, including chemistry, molecular biology, and preclinical research.” So it should not come as a surprise that if one of us uses a specific method, it is mentioned as one of the possible options. Also in this particular case, the publication from which the procedure comes was quoted.

  1. Regarding Determining Partition Coefficient (Log P), Distribution Coefficient (Log D) and Ionization Constant (pKa) in Early Drug Discovery, of course you can recommend HPLC, but not every laboratory has it, it is an expensive method. For exploratory or proof-of-principle work, you can do without this. It is clear that when a drug is at the preclinical testing stage, it will be necessary to fully characterize the drug, including HPLC.

We are fully aware of the fact that not all laboratories are equipped with devices enabling all basic tests recommended in this study to be performed. This does not apply only to HPLC instruments, e.g. Reviewer 1 recommends using a flow cytometer to determine cytotoxicity. At the end of the guide, where we included a list of example laboratories where such measurements can be performed, we wrote "The boron agents are synthesized in chemistry laboratories and not all laboratories are equipped to perform the proposed tests, to facilitate the early bioprofiling procedure we include a sample list of academic laboratories where such tests can be performed as a service".

  1. 2.1.4. Evaluation of the compound's stability at various pH and in human plasma, 2.2.4. Stability of the compounds in human plasma. The same thing - there is a classic, widespread, cheaper method for determining nuclease activity using electrophoretic analysis.

This comment is not clear. Organoboron and, more broadly, organic compounds are degraded under the influence of many factors, both chemical and biological. In a biological environment not only chemical but also enzymatic degradation takes place. Nucleases, however, are specific enzymes and are responsible for cleaving the phosphodiester bonds between nucleotides of nucleic acids. Most organoboron compounds, unless they are not boron-modified DNA/RNA-oligonucleotides, are not substrates for these enzymes.

  1. 2.1.5. Cellular toxicity. First, please explain why these particular cell lines - for example, T98G, and not U87MG, not U343, not U251, etc. (https://www.atcc.org/search#q=glioblastoma&sort=relevancy&numberOfResults=24). Why such a sarcoma (https://www.atcc.org/search#q=squamous%20cell%20line&sort=relevancy&numberOfResults=24).

Why this particular control? (https://www.atcc.org/search#q=Vero&sort=relevancy&numberOfResults=24). These cells are of mesenchymal origin, but the main thing is that it is obtained from the epithelium of the kidney taken from the African green monkey (Chlorocebus aethiops), that is, not a person! Secondly, refer to generally accepted methods - (Assay Guidance Manual. Markossian S, Grossman A, Brimacombe K, et al., editors. Bethesda (MD): Eli Lilly & Company and the National Center for Advancing Translational Sciences; 2004), which does not describe the use of neutral red. Judging by your protocol, you recommend selecting apoptotic cells detached from the plastic, leaving only living ones attached to the surface of the plastic, and staining the lysosomes? Which is not entirely correct, since you are taking some of the cells, usually a tetrazolium salt preparation is dripped on top, without affecting the cells in the wells of the plate. Please provide a link to a study that shows the absence of interaction of eurhodin dye with boron clusters, similar to how you write about MTT (“However, the redox activity of some boron carriers, especially those with a boron cluster in their structure, may overlap with the process of MTT reduction to formazan, thus distorting the readings of cell viability").

Thank you for the comment and discussion. Concerning the question “why these particular cell lines”? Boron compounds are tested in different cell lines, including B16, B16F10, MKT, A1059, TA1059-1, MRA 27, SK-23 Mel, C6, U87-MG, LN-229, T98G, KB, FaDu, SAS, A-253, HuH-7, and many others. Testing boron compounds in all possible cell lines is not justified. In the early stages of boron compound evaluation, a few lines are enough, so one has to choose. In explanation of our choice, we wrote: “As basic lines for preliminary cellular cytotoxicity tests, we propose two cell lines, human glioblastoma multiforme cells T98G and squamous cell carcinoma, FaDu, as an example of cancer cells, and Vero cell line (CCL-81, Cercopithecus aethiops normal kidney cells) is an example of normal tissue cells.” The lines T98G and FaDU represent the two main types of cancer of interest for BNCT.

Concerning a request “Please provide a link to a study that shows the absence of interaction of eurhodin dye with boron clusters, similar to how you write about MTT (“However, the redox activity of some boron carriers, especially those with a boron cluster in their structure, may overlap with the process of MTT reduction to formazan, thus distorting the readings of cell viability").

The redox activity of boron clusters is known and common, and this fact alone prompts caution in the use of colorimetric tests based on redox reactions, as in the case of MTT test based on mitochondrial oxidoreductase activity. The neutral red incorporation test (NR), in turn, is independent of metabolism and is based on the preferential incorporation of the dye into lysosomes.

Although we have encountered the above-mentioned problem for a long time, we recently described the comparison of the results of cytotoxicity tests using the MTT method and the use of neutral red. Our data obtained for nucleoside conjugates with metallacarboranes and unmodified metallacarboranes showed that the compounds caused an overproduction of formazan inside the cell regardless of influence on cell viability, evaluated by ATP or NR assays. The results of MTT test were therefore inconsistent with other viability assays (Cancers 2021, 13, 3855; https://doi.org/10.3390/ cancers13153855 and supplementary section). More detailed studies of this phenomenon are described in the further work submitted for publication.

  1. Regarding 2.1.6. Cellular uptake, then both methods (ICP-MS, ICP-AES) are suitable, the main thing is to indicate the sensitivity of the device, but for some reason at some point you lost the mention of ICP-MS (L.128). Add please.

 The statement “It's worth remembering that ICP-AES provides a higher detection limit down to ppm or ppb, whereas ICP-MS provides a lower detection limit down to ppt” was aded.

  1. For some reason, your pipeline does not include an assessment of the cytotoxicity of BNCT against cells using the developed drugs - a clonogenic test (doi: 10.1038/nprot.2006.339) or RTCA xCelligence. It is worth noting that here, after irradiation, the MTT test is just not suitable (DOI:10.1038/s41598-018-19930-w)!

Thank you for bringing our attention to clonogenic assays useful in determining cell death after irradiation. We limit the proposal to the first, basic stage of evaluation of boron compounds and assume that in vitro neutron irradiation experiments along with basic in vivo tests will be carried out in the next stage on compounds that have successfully passed the initial selection. The cited guide by Hattori and colleagues (ref. 11) can be treated as a valuable guideline for such more advanced research.

  1. Further, after reading the last section, I was perplexed - is this a commercially commissioned article, an advertisement? Why can’t these tests be done anywhere else? After reading your list, how would researchers feel who have three ICP-AES within walking distance but are not listed on your list? Another point, maybe this is a list of laboratories affiliated with the Ministry of Health for performing preclinical tests and they are all done according to GFP standards? Explain what their preference is? They should be removed from the article altogether, since this is advertising!

We can assure the reviewer that the proposed guide is not a commissioned work and the laboratories mentioned in the list are not included there for advertising purposes. Both Reviewers 1 and 2 regrets that not all laboratories are equipped with the equipment needed to perform the recommended in-guide tests (HPLC or flow cytometer, are examples). We would also like to stress that the list includes only academic institutions, no companies providing this type of service are mentioned. Since we are not able to make a complete list of all academic laboratories of this type around the world, only a few examples in each category are provided. The list is quite random, there is no obstacle to adding laboratories indicated by the reviewer to the list if they agree. Additionally, we would like to stress that there is no justification to the question "Why can't these tests be done anywhere else?". It is not clear to us on the basis of which considerations the reviewer raises this question. The explanation for our approach can be found in the first sentence of the appendix: “The boron agents are synthesized in chemistry laboratories, and not all laboratories are equipped to perform the proposed tests, to facilitate the early profiling procedure we include a sample list of academic laboratories where such tests can be performed as a service”.

Round 2

Reviewer 2 Report

Comments and Suggestions for Authors

Dear authors,

You partially answered my questions; however, since I did not receive a specific answer to some of them, I still cannot unambiguously set the status of the article as accepted.

Once again, a perfect example of this kind of article is the MIQE guidelines (Bustin SA, Benes V, Garson JA, Hellemans J, Huggett J, Kubista M, Mueller R, Nolan T, Pfaffl MW, Shipley GL, Vandesompele J, Wittwer CT. The MIQE guidelines: minimum information for publication of quantitative real-time PCR experiments. Clin Chem. 2009 Apr;55(4):611-22. doi: 10.1373/clinchem.2008.112797. Epub 2009 Feb 26. PMID: 19246619). It is informative, written quite clearly, which is actually reflected in a huge number of citations. It is worth noting that there is already an article devoted to BNCT (Yoshihide Hattori, Tooru Andoh, Shinji Kawabata, Naonori Hu, Hiroyuki Michiue, Hiroyuki Nakamura, Takahiro Nomoto, Minoru Suzuki, Takushi Takata, Hiroki Tanaka, Tsubasa Watanabe, Koji Ono, Proposal of recommended experimental protocols for in vitro and in vivo evaluation methods of boron agents for neutron capture therapy, Journal of Radiation Research, Volume 64, Issue 6, November 2023, Pages 859–869, https://doi.org/10.1093/jrr/rrad064), so it makes sense to look at it for the absence of possible overlaps.

My comments to your answers:

Question 1. If you write that "the nanoparticle and polymeric boron carriers deserve a separate guide", then you should change the title of the manuscript to clarify that this manual is intended only for low-molecular-weight organoboron compounds.

Question 2. You did not answer my specific question; you provide links to two reviews. A link to the actual experiment described in the experimental article should be provided.

Question №3. I still think that briefly, “separated by commas,” it would be appropriate to list the most promising ones here. Otherwise, it seems that you are biased in describing only these boron delivery agents.

Question №4. The question has been removed. Now the text looks much better.

Question № 5. You have moved the image to the Introduction, but the resolution is very poor, the text, especially on the right, is impossible to read.

Question № 6. You are absolutely right that "Most journals require detailed information in the experimental section about the equipment used in specific measurements, including the type and name of the instrument and the name of the supplier, so we have followed this format», however, it is appropriate to supplement this information with the physical basis of the method, give an example of your device, and give the reader the opportunity to decide which nephelometer he should buy for work. In addition, this is not a section on materials and methods, where you need to briefly give the name of the device, the name of the company - manufacturer and country.

Question № 7. This is an international scientific journal, and relying on the argument that “The selection of tests and associated methods is based on the practical experience of the authors who have worked for many years in various fields of BNCT” is nonsense. The argument can only be experiment, proven and published facts. Moreover, this sentence should have been removed from the annotation altogether; it is not modest.

Question № 8. Thank you for your answer.

Question № 9. Regarding plasma you write, see below («An example procedure for testing the stability of compounds in human plasma is shown 262 in the experimental section below»). Please provide a specific link/reference - specifically what method you recommend.

Question №10. It would be better to write here that you propose to use two cell lines - one glioblastoma cell line (for example, U87MG, U343 MG, U251 MG) and one carcinoma (for example, FADU, NCI-H520, SiHa etc.), since BNCT is often aimed at treating gliomas and head and neck tumors. And you should definitely bring a few, but not T98G!  Because «U87MG cells (wild-type p53) are known to be tumorigenic in nude mice, but T98G cells (mutant p53) are not tumorigenic (doi: 10.1186/1477-5956-10-53. Typically, if you get excellent results in a cell model, you move to an animal model and use the same cells to create xenografts. This is done for the purity of the experiment. As for VERO cells, their choice is very questionable, since these cells are obtained from the epithelium of the kidney taken from the African green monkey (Chlorocebus aethiops), not humans! There are special sites and catalogs where you can select the appropriate tumor and normal human cell lines (https://www.proteinatlas.org/humanproteome/cell+line/Brain+cancer , https://www.atcc.org/search#q=Squamous%20Cell%20Carcinoma&sort=relevancy&numberOfResults=24&f:Productcategory=[Human%20cells]&f:Disease=[Squamous%20Cell%20Carcinoma] , https://www.cellosaurus.org/search?query=Normal+cell+lines+human).

Regarding your data on « Our data obtained for nucleoside conjugates with metallacarboranes and unmodified metallacarboranes showed that the compounds caused an overproduction of formazan inside the cell regardless of influence on cell viability, evaluated by ATP or NR assays», please provide a link to the published data or provide experimental data in the response to the reviewer, in the supplementary files. Since after clicking on the link (Cancers 2021, 13, 3855; https://doi.org/10.3390/ cancers13153855 and supplementary section) I read the following information on the page 2: “Our data obtained for all nucleoside conjugates with metallacarboranes and unmodified metallacarboranes showed that the compounds caused an overproduction of formazan inside the cell regardless influence on cell viability, evaluated by ATP or NR assays. The results of MTT test were therefore inconsistent with other viability assays (data not shown)”. I am sure this is very interesting information and it is necessary to provide the reader with experimental data confirming this thesis.

Question № 11. Thank you for your answer.

Question № 12. Thank you for your answer.

Question № 13. I also join my colleagues (reviewers), I also regret that not all laboratories are listed here. But “aut Caesar, aut nihil”. And if so, then it is better to remove them from this subsection, since laboratories of the USA, China, Taiwan, Russia, Argentina, that is, laboratories of countries where there are accelerators and/or reactors, are not represented. So far this subsection is very similar to advertising. Besides that, you didn't answer my question – «Another point, maybe this is a list of laboratories affiliated with the Ministry of Health for performing preclinical tests and they are all done according to GFP standards? Explain what their preference is?»

 The boron agents are synthesized in chemistry laboratories, and not all laboratories are equipped to perform the proposed tests, to facilitate the early profiling procedure we include a sample list of academic laboratories where such tests can be performed as a service”. Your last sentence is an advertisement for these laboratories; they will perform tests on a commercial basis!

Author Response

Subject: Response to Reviewer Comments and Revised Submission for Manuscript ID: cells-2916019

Dear Reviewer,

Thank you for helping us to improve this article. Your obvious interest in this topic drives our efforts forward.

You partially answered my questions; however, since I did not receive a specific answer to some of them, I still cannot unambiguously set the status of the article as accepted.

Thank you for your additional comments and suggestions. We hope that this time we have been able to respond in a way that you can accept.

Once again, a perfect example of this kind of article is the MIQE guidelines (Bustin SA, Benes V, Garson JA, Hellemans J, Huggett J, Kubista M, Mueller R, Nolan T, Pfaffl MW, Shipley GL, Vandesompele J, Wittwer CT. The MIQE guidelines: minimum information for publication of quantitative real-time PCR experiments. Clin Chem. 2009 Apr;55(4):611-22. doi: 10.1373/clinchem.2008.112797. Epub 2009 Feb 26. PMID: 19246619). It is informative, written quite clearly, which is actually reflected in a huge number of citations.

Our response to the reviewer:

In our article real time PCR experiments are not mentioned and we therefore do not see the need to refer to the publication above.

It is worth noting that there is already an article devoted to BNCT (Yoshihide Hattori, Tooru Andoh, Shinji Kawabata, Naonori Hu, Hiroyuki Michiue, Hiroyuki Nakamura, Takahiro Nomoto, Minoru Suzuki, Takushi Takata, Hiroki Tanaka, Tsubasa Watanabe, Koji Ono, Proposal of recommended experimental protocols for in vitro and in vivo evaluation methods of boron agents for neutron capture therapy, Journal of Radiation Research, Volume 64, Issue 6, November 2023, Pages 859–869, https://doi.org/10.1093/jrr/rrad064), so it makes sense to look at it for the absence of possible overlaps.

Our response to the reviewer:

As you will have noticed, one of our co-authors is also a co-author of the article you refer us to (Hiroyuki Nakamura). We are convinced that our contribution is an useful addition to the publication by Hattori et al. which is quoted under ref. 15.

My comments to your answers:

Question 1. If you write that "the nanoparticle and polymeric boron carriers deserve a separate guide", then you should change the title of the manuscript to clarify that this manual is intended only for low-molecular-weight organoboron compounds.

Our response to the reviewer:

We changed the title of our article as follows: Early stage in vitro bioprofiling of potential low-molecular-weight organoboron compounds for Boron Neutron Capture Therapy (BNCT) - Proposal for a guide

Question 2. You did not answer my specific question; you provide links to two reviews. A link to the actual experiment described in the experimental article should be provided.

Our response to the reviewer:

To avoid misunderstandings caused by the use of the term "targeted", we have changed the wording of the sentence “Targeted boron agents (DNA, mitochondria) likely will require lower levels than this.”

It now reads: "If boron agents were developed to target subcellular structures such as DNA, the mitochondrial membrane, etc., they would probably require lower concentrations than these."

The square law of distance also applies at the subcellular level. If the n,alpha reaction from the capture of neutrons by B-10 takes place in the vicinity of a sensitive structure, the probability of hitting it is correspondingly increased, which reduces the total concentration required for an effect.

Question №3. I still think that briefly, “separated by commas,” it would be appropriate to list the most promising ones here. Otherwise, it seems that you are biased in describing only these boron delivery agents.

Our response to the reviewer:

In our opinion, a long list of potential boron compounds for BNCT would not fit the readability of the manuscript and can never be exhaustive. To avoid focusing on 2 promising candidates, we have expanded the corresponding paragraph, which now reads:

In search of better boron carriers than the BPA or BSH used to date [8], new preparations are constantly being proposed, such as the recently described boronotyrosine [9] or pteroyl-closo-dodecaborate (PBC-IP) conjugated with a 4-(p-iodophenyl)butyric acid moiety [10], as well as the meta isomer of L-BPA, (L)-3-dihydroxy-borylphenylalanine or 3-BPA[11] that has a better solubility as compared to L-BPA and may result interesting innovative fomulations. Lists of other candidates for boron compounds for BNCT can be found in a recently published article by Oloo et al. [12] and on the website of the German BNCT Society DGBNCT [13]. Special attention merits the need to develop a theranostics approach for BNCT [14].

Question №4. The question has been removed. Now the text looks much better.

We agree.

Question № 5. You have moved the image to the Introduction, but the resolution is very poor, the text, especially on the right, is impossible to read.

Our response to the reviewer:

Thank you, we could not see this error in our original manuscript. It has now been corrected. We also increased the font size to further improve readability.

Question № 6. You are absolutely right that "Most journals require detailed information in the experimental section about the equipment used in specific measurements, including the type and name of the instrument and the name of the supplier, so we have followed this format», however, it is appropriate to supplement this information with the physical basis of the method, give an example of your device, and give the reader the opportunity to decide which nephelometer he should buy for work. In addition, this is not a section on materials and methods, where you need to briefly give the name of the device, the name of the company - manufacturer and country.

Our response to the reviewer:

We deleted the words “NEPHELOstar Plus (BMG LABTECH)”.

Question № 7. This is an international scientific journal, and relying on the argument that “The selection of tests and associated methods is based on the practical experience of the authors who have worked for many years in various fields of BNCT” is nonsense. The argument can only be experiment, proven and published facts. Moreover, this sentence should have been removed from the annotation altogether; it is not modest.

Our response to the reviewer:

We changed the sentence you criticize and it now reads:

The selection of assays and corresponding methods is based on the practical experience of the authors and is certainly not exhaustive, but open to discussion.

Question № 8. Thank you for your answer.

Question № 9. Regarding plasma you write, see below («An example procedure for testing the stability of compounds in human plasma is shown in the experimental section below»). Please provide a specific link/reference - specifically what method you recommend.

Our response to the reviewer:

A reference 26 and 27 is provided in the Section 2.2.4, they are also provided now just after the above-mentioned statement. 

Question №10. It would be better to write here that you propose to use two cell lines - one glioblastoma cell line (for example, U87MG, U343 MG, U251 MG) and one carcinoma (for example, FADU, NCI-H520, SiHa etc.), since BNCT is often aimed at treating gliomas and head and neck tumors. And you should definitely bring a few, but not T98G!  Because «U87MG cells (wild-type p53) are known to be tumorigenic in nude mice, but T98G cells (mutant p53) are not tumorigenic (doi: 10.1186/1477-5956-10-53)». Typically, if you get excellent results in a cell model, you move to an animal model and use the same cells to create xenografts. This is done for the purity of the experiment. As for VERO cells, their choice is very questionable, since these cells are obtained from the epithelium of the kidney taken from the African green monkey (Chlorocebus aethiops), not humans! There are special sites and catalogs where you can select the appropriate tumor and normal human cell lines (https://www.proteinatlas.org/humanproteome/cell+line/Brain+cancer , https://www.atcc.org/search#q=Squamous%20Cell%20Carcinoma&sort=relevancy&numberOfResults=24&f:Productcategory=[Human%20cells]&f:Disease=[Squamous%20Cell%20Carcinoma] , https://www.cellosaurus.org/search?query=Normal+cell+lines+human).

Our response to the reviewer:

Thank you for insisting on this important issue that we have not paid enough attention to. After discussion with colleagues in Japan and Europe, we have amended the text as follows:

As basic lines for preliminary cellular cytotoxicity tests, it is advisable to choose 2 or 3 tumor cell lines, which show good colony formation potentials, good tumorigenic potential in mouse models, without showing elevated multidrug resistance (MDR) pumps expression. For BNCT experiments, SAS, U87MG, and A375 can be recommended for oral cancer, high grade glioma, and melanoma, respectively. Normal cell lines can be used as controls for cell-based studies if necessary. HEK293 can be utilized for "normal" kidney cells. They are not quite "normal" as they are transfected with SV40 to be immortal and form tumors under the skin in mice. But in vitro this is an acceptable model.  

The sections in the manuscript where reference is made to the cell lines and related publications have been adapted accordingly.

Regarding your data on « Our data obtained for nucleoside conjugates with metallacarboranes and unmodified metallacarboranes showed that the compounds caused an overproduction of formazan inside the cell regardless of influence on cell viability, evaluated by ATP or NR assays», please provide a link to the published data or provide experimental data in the response to the reviewer, in the supplementary files. Since after clicking on the link (Cancers 2021, 13, 3855; https://doi.org/10.3390/ cancers13153855 and supplementary section) I read the following information on the page 2: “Our data obtained for all nucleoside conjugates with metallacarboranes and unmodified metallacarboranes showed that the compounds caused an overproduction of formazan inside the cell regardless influence on cell viability, evaluated by ATP or NR assays. The results of MTT test were therefore inconsistent with other viability assays (data not shown)”. I am sure this is very interesting information and it is necessary to provide the reader with experimental data confirming this thesis.

Our response to the reviewer:

As we explained in the previous response to the reviewer, the full paper in which we discuss, among others, the problem of overlapping the redox activity of some boron carriers with the process of MTT reduction to formazan, is under review and we are unable to fulfill the reviewer's request to provide the reader with experimental data confirming this observation and disclose unpublished results. Therefore, with regret we must remove the statement “However, the redox activity of some boron carriers, especially those with a boron cluster in their structure, may overlap with the process of MTT reduction to formazan, thus distorting the readings of cell viability” from the present version of the manuscript, although it might be of interest for the readers. We would also be grateful if the reviewer treated this information as confidential until the paper was accepted for publication.

Question № 11. Thank you for your answer.

Question № 12. Thank you for your answer.

Question № 13. I also join my colleagues (reviewers), I also regret that not all laboratories are listed here. But “aut Caesar, aut nihil”. And if so, then it is better to remove them from this subsection, since laboratories of the USA, China, Taiwan, Russia, Argentina, that is, laboratories of countries where there are accelerators and/or reactors, are not represented. So far this subsection is very similar to advertising. Besides that, you didn't answer my question – «Another point, maybe this is a list of laboratories affiliated with the Ministry of Health for performing preclinical tests and they are all done according to GFP standards? Explain what their preference is?»

Our response to the reviewer:

The list of laboratories has been obtained by asking colleagues, who own the methods, and which are willing to give help and some support to colleagues. These laboratories are not affiliated with a Ministry of Health and they are mostly not following GFP standards.

 “The boron agents are synthesized in chemistry laboratories, and not all laboratories are equipped to perform the proposed tests, to facilitate the early profiling procedure we include a sample list of academic laboratories where such tests can be performed as a service”. Your last sentence is an advertisement for these laboratories; they will perform tests on a commercial basis!

Our response to the reviewer:

We agree that such a list of laboratories that have agreed to support other academic institutions can be seen as advertising. We have tried to exclude this aspect by only listing academic institutions that have no commercial interest in offering such a service. As we know that many of our colleagues are dependent on such support, we do not wish to abandon this list. We hope that this will provide useful support for the necessary basic research in the development of new boron carriers for BNCTs and open up the possibility of working in this field to laboratories that have not previously dealt with this topic. We change the sentence you criticize by deleting the wording “as a service”. I now reads as follows:

The boron agents are synthesized in chemistry laboratories, and not all laboratories are equipped to perform the proposed tests, to facilitate the early profiling procedure we include a sample list of academic laboratories where such tests can be performed and which can give some support.

We also changed the title of this chapter as follows:

List of example laboratories where preliminary bioprofiling tests of potential boron carriers can be carried out

We believe that with these revisions, our manuscript is now well-prepared for publication in Cells. Should you require any further information for the final acceptance of our manuscript, please do not hesitate to contact us.

In any case, thank you very much for your extremely important support!

Wolfgang Sauerwein and Zbigniew Lesnikowski
